# Event-based backpropagation on the neuromorphic platform SpiNNaker2

**Gabriel Béna**
SpiNNcloud Systems, Dresden, Germany
`g.bena@spinncloud.com`

**Timo Wunderlich**
Universitätsmedizin Berlin, Germany

**Mahmoud Akl**
SpiNNcloud Systems, Dresden, Germany

**Bernhard Vogginger**
TU Dresden, Germany

**Christian Mayr**
TU Dresden, Germany
ScaDS.AI Dresden/Leipzig, Germany

**Hector Andres Gonzalez**
SpiNNcloud Systems, Dresden, Germany
TU Dresden, Germany
ScaDS.AI Dresden/Leipzig, Germany

## Abstract

Neuromorphic computing aims to replicate the brain's capabilities for energy efficient and parallel information processing, promising a solution to the increasing demand for faster and more efficient computational systems. Efficient training of neural networks on neuromorphic hardware requires the development of training algorithms that retain the sparsity of spike-based communication during training. Here, we report on the first implementation of event-based backpropagation on the SpiNNaker2 neuromorphic hardware platform. We use EventProp, an algorithm for event-based backpropagation in spiking neural networks, to compute exact gradients using sparse communication of error signals between neurons. Our implementation computes multi-layer networks of leaky integrate-and-fire neurons using discretized versions of the differential equations and their adjoints, and uses event packets to transmit spikes and error signals between network layers. We demonstrate a proof-of-concept of batch-parallelized, on-chip training of spiking neural networks using the Yin Yang dataset, and provide an off-chip implementation for efficient prototyping, hyper-parameter search, and hybrid training methods.

38th Second Workshop on Machine Learning with New Compute Paradigms at NeurIPS 2024(MLNCP 2024).

# 1 Introduction

Neuromorphic computing seeks to emulate the unparalleled efficiency of biological neural networks by implementing spiking neural networks which use sparse, spike-based communication between neurons. This could enable artificial neuronal systems to process temporal, spike-based data with efficiency similar to biological brains. At the same time, backpropagation proved to be a pivotal method in machine learning, allowing for efficient gradient computation and enabling the recent advances in training non-spiking, artificial neural networks on challenging tasks [8]. This suggests that implementing gradient-based learning algorithms on neuromorphic hardware could enable similar achievements while surpassing traditional hardware in terms of energy efficiency. Such learning algorithms should make use of the temporal sparsity afforded by spike-based processing.

Building on previous work on gradient-based learning in spiking neural networks [3], the Event-Prop algorithm [26] uses event-based backpropagation to compute exact gradients, retaining the advantages of temporal sparsity during network training. In contrast, non-spiking neural networks typically require a dense sampling of neuronal variables for backpropagation which introduces a memory bottleneck, limits network size and increases energy consumption due to the required storage and transfer of dense state variable samples [20, 16, 12]. Harnessing the potential of event-based algorithms such as EventProp requires their implementation on suitable event-based hardware architectures.

Gradient-based learning using neuromorphic hardware has been previously implemented using "in-the-loop" approaches, where neuromorphic hardware computes spike times and state variables in an inference phase (the forward pass) while a classical computer computes gradients and implements backpropagation for the computation of surrogate gradients [4, 5] or exact gradients [6, 18]. Previous work implemented the E-prop algorithm [1] on SpiNNaker1 [19] and an FPGA-based prototype of SpiNNaker2 [21]. This algorithm computes surrogate gradients using local eligibility traces and a global error signal, without error backpropagation through network layers, which prevents scalability. Besides SpiNNaker, the GeNN code generation framework [10] has been used to create a GPU-based implementation of EventProp [15].

This manuscript presents the first implementation of event-based backpropagation on SpiNNaker2. SpiNNaker2 is a novel digital neuromorphic hardware platform providing a scalable event-based and asynchronous computational substrate [7]. While our results are obtained using a single SpiNNaker2 chip, the platform can scale to large numbers of interconnected chips (e.g., more than 5 million compute cores in the Dresden SpiNNcloud platform [13]).

# 2 Methods

## 2.1 SpiNNaker2 Details

SpiNNaker2 is a massively parallel compute architecture for event-based computing. It is based on a digital, many-core GALS (Globally Asynchronous Locally Synchronous) configuration, where every chip is composed of 152 ARM-based Processing Elements (PEs) with dedicated accelerators for both neuromorphic (e.g., true random number generators, exponential and logarithmic accelerators) and deep learning applications (e.g., multiply-and-accumulate arrays). The PEs are arranged in a two-dimensional array and communicate via a dedicated high-speed Network-On-Chip (NoC). Communication to a host computer is established via $1\,\mathrm{Gbit}$ ethernet (UDP) and $2\,\mathrm{GB}$ of DRAM on the board can be accessed via two LPDDR4 interfaces. For scalable, system-wide communication, each chip has a dedicated SpiNNaker2 packet router, containing configurable routing tables, and six links to neighbouring chips. Different packet types with up to 128-bit payload allow for efficient communication between PEs, chips and boards. This will allow for SpiNNaker2 systems with up to $65\,000$ chips and over 10 million cores [13]. Importantly, these ARM Cortex M4F cores provide flexibility for the implementation of arbitrary neuron dynamics and event-based models (e.g., [23]).

A Python-based software package, PY-SPINNAKER2 [25], allows user to define network architectures and parameters (py-spinnaker2) and is the main entry point for running experiments on-chip. The Python module interacts with the chip through an intermediate layer implemented in C++ that sends and receives data and loads up PEs with the memory files needed to run simulations. Interactions between host and chip are detailed in Figure 1.

## 2.2 Programs on Processing Elements

Our implementation comprises four programs running on different PEs of a chip: the first injects input spikes, the second simulates a layer of leaky integrate-and-fire neurons, the third computes a time-to-first-spike loss and the fourth accumulates gradients and computes weight updates.

A common regularization method in machine learning is to process a subset of the training batch in parallel: gradients are averaged across a *mini-batch* [8]. Mini-batch training can achieve better generalization while speeding up training by processing training examples in a given mini-batch in parallel. In the case of training on Spinnaker2, we can take advantage of the inherently parallel nature of the architecture to deploy identical copies of the network on multiple cores. Different copies of a network process different input samples, implementing mini-batch training. Except for the optimisation program, all described programs are duplicated on the chip for each training example processed in parallel.

We detail each program's implementation next, and provide pseudo-code for each in 6.4. The architecture and software stack is summarised in Figure 1. This implementation showcases the flexibility of SpiNNaker2 to implement custom training algorithms on neuromorphic hardware. While our implementations only uses $128\,\text{kB}$ of SRAM available on each PE, future work will interface with $2\,\text{GB}$ of DRAM on the board to enable larger networks and more complex models.

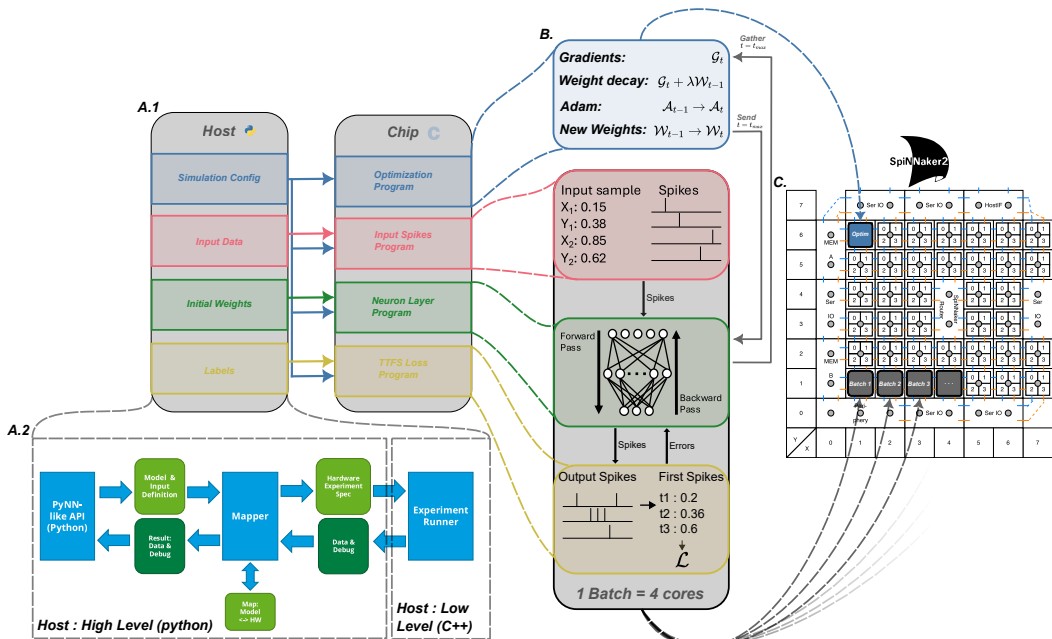

Figure 1: Software and Hardware stack: **A.1** shows the software architecture stack. The host experiment runner handles network creation, simulation parameters, initial data and mapping, all of which are detailed in **A.2**. Necessary data is then sent on-chip. **B.** details the 4 main programs responsible for executing the eventprop algorithm on-chip, detailed later on. **C.** shows a potential mapping of a multi-batch simulation on the SpiNNaker 2 chip.

**Input Spikes Program**: This program injects input spikes representing samples of the input data set into neurons of the first layer. We use a dedicated PE running this program to inject spikes into the network for clarity and to optimise memory usage. Spike times for every sample are loaded at the beginning of the simulation and injected using event packets via the network-on-chip to PEs representing target layers. To process different samples of a training mini-batch in parallel, different cores run the same program, but transmit different input samples to neuronal input layers during the simulation.

**Neuron Layer Program**: This program simulates a layer of leaky integrate-and-fire neurons by computing the forward and backward pass. It uses a clock-driven simulation based on the discretized differential equations and their adjoints [26]. In a given time step, different network layers are

computed asynchronously on different PEs and event packets representing spikes (forward pass) or error signals (backward pass) are transmitted by the network-on-chip and buffered by the receiving PEs to update variables in the next time step. PEs store dense weight matrices (in contrast to the original synapse-based implementation of SpiNNaker) and use incoming spikes' payloads to retrieve the synaptic weight corresponding to the arriving spike and to update the synaptic current of the target neuron.

During the backward pass, spikes are distributed in reverse order, and carry state variables representing error signals within the $32 \, \mathrm{bit}$ payload of event packets. At this point, the network runs in "reverse-time" and follows the *adjoint dynamics*, with error signals being transmitted at the spike times computed during the forward pass. In this way, the backward pass implements a spiking neural network with graded spikes (representing the error signal) at fixed times, making it suited for an event-based implementation on the neuromorphic platform. The backward pass implements a discretization of the adjoint system [26], following an "optimize-then-discretize" approach. In this way, the program uses event-based backpropagation to compute a discretized approximation to the exact gradients of the simulated spiking neural network. The complete set of discretized equations is provided in section 6.1.

**Time-to-First-Spike Loss Program**: The loss program receives spikes from the output layer and computes the loss as well as the initial error signals sent to the output layer during the backward pass. At the beginning of the simulation, cores running the loss program are loaded with classification labels. After receiving spikes from the output layers, loss cores compute derivatives of a cross-entropy loss based on first spike times as given by

$$\mathcal{L} = -\log \left( \frac{\exp\left(-(t_l^{\mathrm{s}} \Delta t)/\tau_0\right)}{\sum_{k=1}^{3} \exp\left(-(t_k^{\mathrm{s}} \Delta t)/\tau_0\right)} \right) - \alpha \left[ \exp\left( \frac{t_l^{\mathrm{s}} \Delta t}{\tau_1} \right) - 1 \right], \tag{1}$$

where $t_k^{\mathrm{s}}$ is time step of the first spike of the $k$th neuron, $l$ is the index of the neuron that should fire first and $\tau_0$, $\tau_1$ and $\alpha$ are hyper-parameters of the loss function. Loss cores compute error signals corresponding to the derivative of eq. (1) which are sent back to the output layer at the respective spike time steps using spike payloads.

**Optimisation Program**: After the backward pass, each neuron layer has accumulated gradients. A single PE is then tasked with gathering gradients and optimisation of the weights. The PE running this program gathers gradients using direct memory access (DMA) from PEs processing different samples of a batch and sums them locally, implementing a mini-batch gradient update. The summed gradients are used to update an internal state which implements the Adam optimization algorithm [9]. This state is used to compute a new set of weights which is written back to each PE using DMA. The processing of the next sample is then triggered for all cores using a global interrupt signal, synchronizing the processing elements.

## 2.3 Yin Yang Dataset

We consider a learning task using the Yin Yang dataset [11]. This dataset consists of three classes of points in a two-dimensional space ($[0, 1]^2$), resembling a Yin Yang symbol (Figure 2). Importantly, the dataset is not linearly separable and designed to be challenging for linear classifiers without any hidden layers, requiring deep networks for effective classification (a shallow classifier achieves around $64\%$ accuracy [11]). We use this dataset due to the current memory limitation of our implementation and the clear performance gap between shallow and deep networks, allowing us to validate that our implementation can train a deep network on non linear-separable data. We encode the dataset using spikes by translating each coordinate into a spike time in the interval from $t_{\min} = 2$ to $t_{\max} = 27$, running the networks for a total of $T = 28$ timesteps. Following [11], we add coordinates $(1 - x, 1 - y)$ for each data point $(x, y)$. We also add a bias spike at time $t = 0$, resulting in a total of 5 input neurons. Spike times are discretized according to the simulation time step, with ambiguous data points being removed (Figure 2, right plot).

## 2.4 Off-chip simulation

In addition to the on-chip implementation of the EventProp algorithm, we developed an off-chip simulation that matches the on-chip simulation and can be used for efficient prototyping, convenient debugging and parameter search. This off-chip implementation takes the form of a custom PyTorch

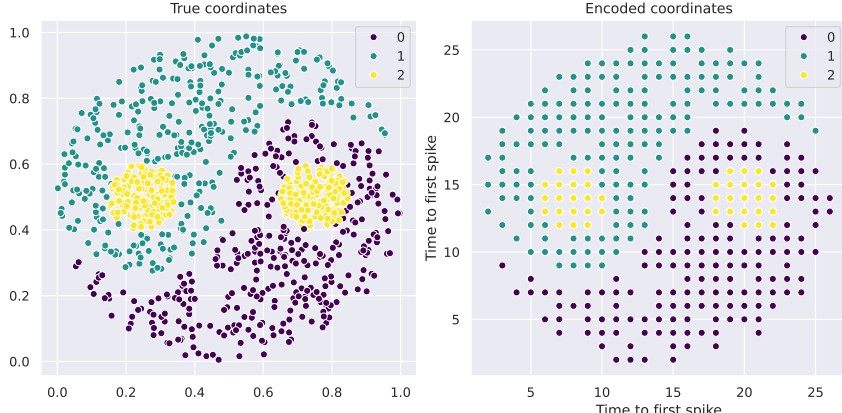

Figure 2: Yin Yang Dataset: Left-hand side shows true coordinates coloured by labels, while right-hand side shows the time steps of discretized spike-times.

[17] package, and is available at pytorch-eventprop. A quantitative comparison of the on- and off-chip simulations is provided in Figure 4, Figure 5 and shows a minimal mismatch between state variables that is expected due to numerical differences. Future work could leverage the off-chip simulation for a hybrid training approach that trains a base model off-chip and then deploys it to SpiNNaker2 for on-chip adaptation.

We use the off-chip simulation to perform a Bayesian hyperparameter optimization using the "Weights and Biases" software package [2]. The resulting parameters are given in Table 2.

## 3  Results

**Yin Yang Dataset**

We present learning results using the Yin-Yang dataset and network's dimensions $[5, 120, 3]$. Total training-set size is $5000$, processing in parallel mini-batches of size $22$ . We train for 40 epochs in total, and we average results over 10 different random seeds (Figure 3, left). These results demonstrate successful on-chip training using event-based backpropagation on SpiNNaker2 as well as a close match between on- and off-chip simulations after training.

In addition to batch-parallelised processing, we report on results where the networks are trained on a single sample at a time, for a single epoch, on a limited training-set of $300$ samples (Figure 3, right). Such a scenario is particularly relevant for on-line learning on neuromorphic hardware, where the network is trained on a continuous stream of data, which would be critical for embedded systems and autonomous agents. Again, the results show a close match between on-chip and off-chip simulations, and satisfying accuracy considering the heavy limitations.

**Profiling**

We conducted performance analysis comparing GPU-based off-chip and Spinnaker implementations based on processing time and energy consumption for single batch computation. Using 30-millisecond input samples, the GPU-based implementation demonstrated significant speed advantages, achieving approximately 10x acceleration compared to real-time processing. The Spinnaker implementation maintained real-time processing capabilities, successfully processing samples within the 61ms (forward + backward + 1 optimization step) window. Power efficiency measurements revealed substantial differences between the implementations. The Spinnaker implementation exhibited minimal power consumption of under 0.5W, while power measurements for the GPU-based implementation indicated a consumption of 13.5 W for the GPU device alone. It is important to note that this measurement excludes the power consumption of other system components such as the CPU, memory, and peripheral devices, suggesting that the total system power draw would be notably higher. These GPU measurements warrant additional careful interpretation due to

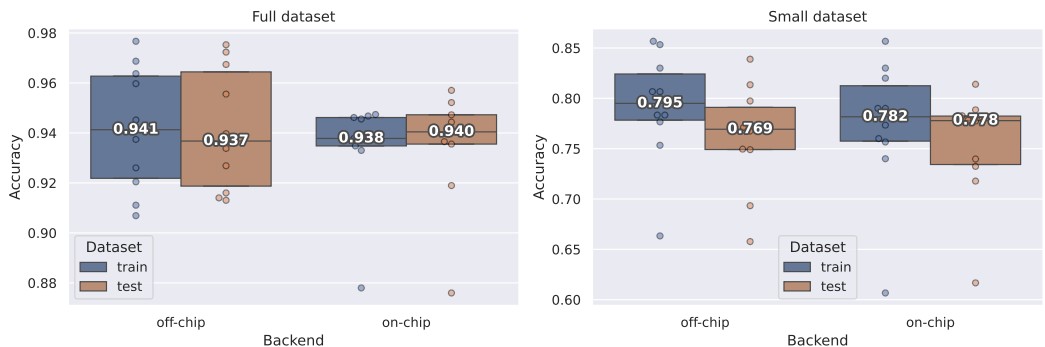

Figure 3: Accuracy comparison between on- and off-chip simulations. On the left, we show the final accuracy of the models after training for 40 epochs on the complete dataset, for both testing and training sets. On the right we show the final accuracies reached in the "online" setting, after only seeing 300 samples one by one. Results are displayed using a standard boxplot, displaying the median (labelled) and quartiles values of the distribution. All individual points are also overlayed on top.

their collection on a shared workstation environment where other processes may have influenced power draw. To establish a more rigorous comparison, future work could leverage embedded GPU platforms such as the Jetson Nano, which would provide isolated testing conditions and precise power monitoring capabilities. Such platforms would allow for measurement of both isolated GPU power consumption and total system power draw, potentially offering a more equitable comparison with the Spinnaker implementation's full-system measurements. While our current measurements suggest superior energy efficiency for the Spinnaker implementation, a comprehensive comparison would require controlled testing environments and dedicated power monitoring infrastructure to draw definitive conclusions about the relative energy efficiency of these approaches. The profiling results are shown in table 1

Table 1: Profiling Results

|  | Time | Energy |
|---|---|---|
| GPU | 6.61 ms ± 427 μs | 13.5 W |
| SpiNNaker 2 | 61 ms | 450 mW |

## 4 Discussion

This work reports on the first, proof-of-concept implementation of event-based backpropagation on SpiNNaker2. While our results are limited by the memory capacity of each processing element, future work will leverage the 2 GB of DRAM available on the host board to enable larger networks and the processing of more complex data sets. However, scaling the current implementation to larger networks and multi-chip systems will require addressing several challenges. For instance, the discretization used in this implementation of EventProp may impose constraints on the scalability and performance of multi-chip systems. Discretization errors accumulate and likely limit the scalability of this approach, especially for deep networks or networks with long temporal dependencies. This challenge could be addressed by using other neuron models and numerical schemes that allow for scalable event-based neural networks.

While our work is based on spiking neurons, its applicability is not confined to this model class. Advances in event-based machine learning, such as the event-based gated recurrent unit [23], suggest the possibility of extending neuromorphic event-based backpropagation to models that use event-based processing with graded spikes. Hybrid hardware such as SpiNNaker2 that supports both event-based communication and the acceleration of conventional deep neural networks could facilitate the integration of dynamic sparsity benefits into broader machine learning applications, leading to order-of-magnitude improvements in energy efficiency compared to traditional computing [14].

Our on-line learning results suggest that event-based backpropagation could be used for on-chip learning in applications where training data arrives continuously. The computational complexity of the backward pass corresponds to that of the forward pass, making it feasible to use EventProp in such a scenario even if the algorithm is not online strictly speaking. Moreover, it would be possible to implement reinforcement learning methods such as policy gradient [24] in an on-line fashion, since EventProp only needs reward signals and not labels. Our framework could use the off-chip simulation to train a base model that is then deployed and fine-tuned on the neuromorphic hardware using data arriving in real time. To achieve efficient and adaptive on-chip fine-tuning across a range of tasks, the off-chip simulation could implement an outer meta-training loop. For example, model agnostic meta learning (MAML) can be used to find an optimal initialisation that ensures quick and effective adaptation when deployed on-chip, and has been shown to be successfully applicable to SNNs [22]. Such a hybrid approach could leverage and combine the advantages of conventional and neuromorphic computing, and enable the deployment of autonomous and adaptive agents on edge-devices.

Our work provides a proof-of-concept implementation of event-based backpropagation using Event-Prop on SpiNNaker2. Event-based backpropagation reduces the demand for memory compared to backpropagation-through-time in non-spiking neural networks and therefore allows for larger network sizes given a fixed amount of memory. At the same time, implementing event-based backpropagation on natively event-based computational substrates such as SpiNNaker2 could enable higher energy efficiency due to temporally sparse data transfers between neurons. Realising these advantages will require addressing the challenges outlined above, and gradient-based learning in spiking neural networks remains an active area of research. Our work demonstrates that the flexibility afforded by SpiNNaker2 can be used to implement future advances in event-based training methods for spiking neural networks.

# 5 Acknowledgement

This project is partially funded by the EIC Transition under the "SpiNNode" project (grant number 101112987), and by the Federal Ministry of Education and Research of Germany in the programme of "Souverän. Digital. Vernetzt.". Joint project 6G-life, project identification number: 16KISK001K. This work was partially funded by the German Federal Ministry of Education and Reseach (BMBF) within the KI-ASIC project (16ES0996), and by the BMBF and the free state of Saxony within the ScaDS.AI center of excellence for AI research. The authors would like to acknowledge the 2022 Telluride Neuromorphic Cognition Engineering Workshop where this project was initiated.

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

# 6 Supplementary Materials

## 6.1 Complete equations for neuron layer program

### 6.1.1 Forward

Denoting the synaptic current and membrane potential of the $j$th neuron in the $l$th layer at timestep $t$ as $I_i^{j,l}$ and $V_i^{j,l}$, our implementation computes:

$$I_{t+1}^{j,l} = \alpha_I I_t^{j,l} + \sum_{k=1}^{N_{l-1}} W_{jk}^l \theta(V_t^{k,l-1} - 1), \tag{2}$$

$$V_{t+1}^{j,l} = \alpha_V V_t^{j,l}(1 - \theta(V_t^{j,l} - 1)) + (1 - \alpha_V)I_{t+1}^{j,l}, \tag{3}$$

where $\alpha_V, \alpha_I \in [0, 1]$ are the respective decay factors, $W_{(jk)}^l$ is the weight matrix of the $l$th layer and $\theta(\cdot)$ is the Heaviside theta function. The number of neurons in the $l$th layer is given by $N_l$ and the initial conditions are $I_0^{j,l} = V_0^{j,l} = 0$ for all neurons. The decay factors are given by

$$\alpha_I = \exp\left(-\frac{\Delta t}{\tau_{\mathrm{s}}}\right), \tag{4}$$

$$\alpha_V = \exp\left(-\frac{\Delta t}{\tau_{\mathrm{m}}}\right), \tag{5}$$

where $\tau_{\mathrm{s}}$ and $\tau_{\mathrm{m}}$ are time constants and $\Delta t$ is the discretization time step.

### 6.1.2 Backward

The backward pass implements a discretization of the adjoint system [26] as

$$\lambda_t^{j,l} = \alpha_I \mu_{t+1}^{j,l} + (1 - \alpha_I)\lambda_{t+1}^{j,l-1}, \tag{6}$$

$$\mu_t^{j,l} = \alpha_V \mu_{t+1}^{j,l} + \theta(V_{t+1}^{j,l} - 1)(I_{t+1}^{j,l} - V_{t+1}^{j,l})^{-1}\left[\mu_{t+1}^{j,l} + \sum_{k=1}^{N_{l+1}} W_{kj}^{l+1}(\mu_{t+1}^{k,l+1} - \lambda_{t+1}^{k,l+1})\right], \tag{7}$$

with initial conditions $\lambda_T^{j,l} = \mu_T^{j,l} = 0$, where $T$ is the final time step.

### 6.1.3 Gradients computation

The gradient matrix is accumulated as

$$G_{jk}^l = -\tau_{\mathrm{s}} \sum_{t=1}^{T} \theta(V_t^{k,l-1} - 1)\lambda_t^{j,l}. \tag{8}$$

## 6.2 Off-chip simulation

Figure 4 shows the differences of voltage traces (forward and backward) between off-chip and on-chip implementations when processing a single sample. The minor deviations, which are likely caused by numerical differences, imply that the off-chip implementation can be used to optimise hyperparameters for on-chip deployment. we can see in Figure 5 that the resulting gradients and weights end up almost perfectly identical, only deviating of minor numerical values.

## 6.3 Optimized Hyperparameters

- **Loss Parameters** : We tune $\alpha$ (regularization strength in the loss function), $\tau_0$ and $\tau_1$ (see 1)
- **Initialisation Parameters**: We tune scales to initialise the neuron populations. By default, the population is initialised following a normal distribution $\mathcal{M}(\mu, \sigma)$, where $\mu = \sigma = \frac{1}{\sqrt{n_{in}}}$, $n_{in}$ being the number of input neurons of the layer. We fine-tune this distribution by scaling $\sigma, \mu$ by factors $s_\sigma^l, s_\mu^l$ for each layer $l$.
- **Optimization Parameters**: We tune lr, $\gamma$ and $\lambda$, the optimizer's learning rate, scheduler's decay rate, and weight-decay parameter respectively

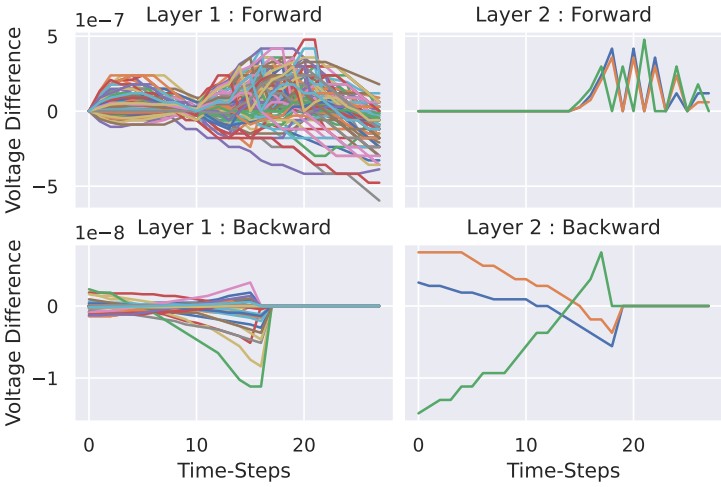

Figure 4: Voltage differences of the on-chip and off-chip implementations after processing a single sample.

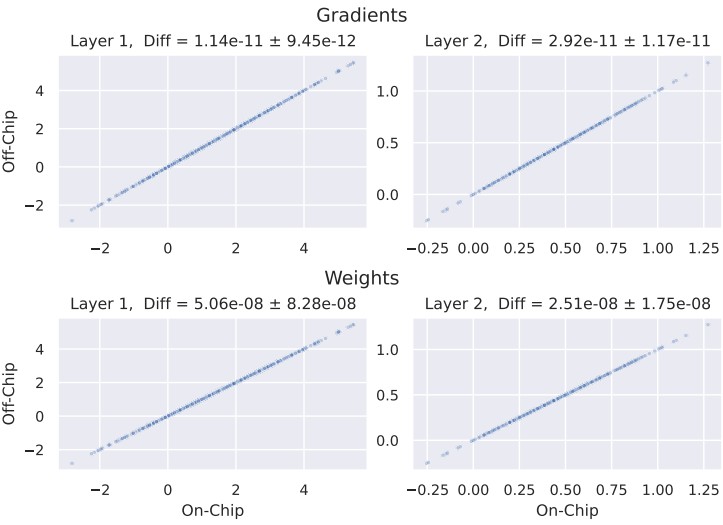

Figure 5: Scatterplot of gradients and weights of the on-chip and off-chip implementations after processing a single sample.

| $\alpha$ | $\tau_0$ | $\tau_1$ | lr | $\gamma$ | $\lambda$ | batch size | T | $(s_\mu^0, s_\sigma^0)$ | $(s_\mu^1, s_\sigma^1)$ |
|---|---|---|---|---|---|---|---|---|---|
| 0.01 | 1.5 | 100 | 0.002 | 0.93 | 6.5e-7 | 22 | 28 | (3.2, 3.2) | (5.2, 2.8) |

Table 2: Best hyper-parameters discovered by Bayesian search

## 6.4   Pseudo Code for Chip Programs

We present here pseudo code for all chip programs to enable readers a clearer idea of the way computation is done on-chip. These should not be taken literally, albeit being close to the actual code they include simplifications for better reading comprehension.

Listing 1: Neuron Population Simulation

```
void reset_and_sync():
    reset_input_buffers();
    reset_neuron_states();
    reset_recordings();
```

```
        // timestep counter reset
        systicks = UINT32_MAX ;
        total_systicks = UINT32_MAX;
        backward = False;
        run = 1;
        // wait for interrupt from control node to ensure optimization is finished
        wait_for_start_signal();
        reset_neuron_gradients();
        timer_init();
        timer_start();

void neuron_process_sample():
    while total_systicks <= n_total_systicks:
        total_systicks++
        if not backward :
            // receive last spikes but do not process
            if systicks == n_timesteps:
                receive_spikes();
            // initialize backward pass
            else if systicks == n_timesteps + 1:
                systicks--;
                backward = True;
                neuron_initialize_backward();
            // Forward Pass
            else:
                systicks++;
                receive_spikes();
                neurons_update();
        // Backward Pass
        else:
            systicks--;
            receive_spikes_backward();
            neuron_do_backward_update();
            accumulate_gradients();
            send_errors_backward();
    run = 0;

while epoch <= n_epochs:
    while sample <= n_samples:
        reset_and_sync();
        while run:
            neuron_process_sample();
        sample++;
    sample = 0;
    epoch++;
```

Listing 2: Loss Population Simulation

```
void reset_and_sync():
    reset_recordings();
    // timestep counter reset
    systicks = UINT32_MAX;
    total_systicks = UINT32_MAX;
    backward = False;
    run = 1;
    // wait for interrupt from control node to ensure optimization is finished
    wait_for_start_signal();
    reset_loss_and_errors();
```

```
        timer_init();
        timer_start();

void loss_process_sample():
    while total_systicks <= n_total_systicks:
        // Prepare for backward
        if not backward:
            if systicks == n_timesteps:
                receive_spikes();
                backward = True;
                systicks--;
                compute_loss();
            // Forward pass
            else:
                systicks++;
                // Receive spikes from output layer
                receive_spikes();
        // Bacward Pass
        else:
            systicks--;
            // Bacpropagate errors at spike time
            send_spikes_backward();
    run = 0;

    while epoch <= n_epochs:
        while sample <= n_samples:
            reset_and_sync();
            while run:
                loss_process_sample();
            sample++;
        sample = 0;
        epoch++;
```

Listing 3: Control Node

```
    while epoch <= n_epochs:
        while sample <= n_samples:
            // Synchronize PEs' start
            send_start_signal();
            // Nothing to do while PEs complete their simulations
            wait_for_completion();
            // Get gradients from all neuron PEs
            gather_gradients();
            // Adam algorithm
            update_adam();
            // Update weights and send back to PEs after simulation
            update_weights();
            sample++;
        sample = 0;
        epoch++;
```

