# OpenReview forum: "Event-based backpropagation on the neuromorphic platform SpiNNaker2"
_NeurIPS.cc/2024/Workshop/MLNCP — MLNCP Poster_

### Official Review · Reviewer_d1Bm · 2024-09-21
**An impressive fully on-chip training SNN experiment... But how much better than feedforward nets training by backprop is it?**

**Rating:** 7
**Confidence:** 4

**Review:**

**Summary.** The present work demonstrates for the first time *on-chip training* of an SNN on the SpiNNaker2 neuromorphic platform using EventProp, a learning technique which doesn't require the storage of the full inference computational graph but only that of the spiking times, yielding important memory savings. Let's emphasize this is *not* an in-the-loop experiment where only inference is performed on the chip: *everything* -- inference, loss computation, backward pass and weight update -- is performed on-chip. The paper demonstrate "successful" training on the Yin-Yang dataset, a 3 class 2d nonlinear classification task in the sense that i) the model performs much better than a shallow classifier, ii) the on-chip experiment matches its "off-chip" counterpart -- where the chip is emulated on standard hardware -- in terms of forward and adjoint voltage alignment (Fig. 4), and therefore of weight gradient alignment (Fig. 5) and of resulting training performance (Fig. 3).
More precisely, the paper describes the SpiNNaker2 high level architecture (2.1), the four types of programs run on the Processing Elements (input processing, forward pass, backward pass, loss computation with Eq. 1, weight updates, 2.2) and how *mini-batching* is achieved (Fig. 1). Then the dataset is introduced (2.3), the idea of benchmarking the hardware experiment with an "off-chip" experiment (2.4) and final results thereof (3, Fig. 3) -- note that an "online setting" with batch size of 1 and fewer training examples is also presented. Then future developments of the work are discussed (4).

**Strengths.**

- The paper is extremely clear, easy to follow and the figures are neat.
- The hardware experiment is the first of its kind on SpiNNaker2, where *everything* is done on chip (including gradient computation and parameter optimization).
- The alignment in terms of forward voltages, adjoint voltages and gradients between off-chip and on-chip experiments is impressive.
- The system allows for mini-batching.

**Weaknesses.**

- I don't understand the relevance of the dataset (static inputs) when using spikes. Why not a *temporal* task?
- When using the full dataset (Fig. 3, left panel), I'm astonished to see a higher variance in resulting performance for the off-chip experiment compared to the on-chip experiment. I would rather have expected the opposite. It would be good to provide some explanation of why this is the case.
- *"Event-based backpropagation reduces the demand for memory compared to backpropagation-through-time in non-spiking neural networks and **therefore allows for larger network sizes given a fixed amount of memory"***. Currently, you are fitting a 5-120-3 net onto the architecture to be trained by EventProp. To further ground the claim about memory savings, it would have been interesting to say which "maximal" architecture size you could have fit onto the *same* chip **if you were using a standard feedforward net trained by standard backprop**. While I do understand why storing the full computational graph of a *convergent* architecture (such as a SNN) would take a lot of memory, is the memory usage of the computational graph of a *feedforward* net so much more important than that of the firing time of the SNN you are training?
- In the light of the previous comment, **I would like to see better benchmarking, on the *same chip*, when training a feedforward net by backprop**.

---

### Official Review · Reviewer_hv2p · 2024-10-04
**A proof-of-concept implementation of event-based backpropagation using Event-Prop on SpiNNaker2**

**Rating:** 7
**Confidence:** 4

**Review:**

The submission provides a proof-of-concept implementation of event-based backpropagation using Event-Prop on SpiNNaker2. The effort is important and relevant for MLNCP as it explores neuromorphic computing as an emerging ML paradigm as well as furthers the pursuit of learning algorithms for neuromorphic. The submission is well written and clear with a base result. Overall, I rate it as a good paper that should be accepted. In the nature of the MLNCP workshop's intention to share work-in-progress, the submission does need additional content to be ready for a formal publication.

A few points of feedback include:
- Figure 1 is very dense and difficult to follow. It seems like some of the terms and software blocks require a deep understand of SpiNNaker2 functionalitiy.
- While the work acknowledges a scaling limitation and the desire to leverage a larger 2GB onboard DRAM, why is it not possible to use a larger example now? Do the 148 unused processing elements not have any memory? Could the parallelization be spread to every PE rather than 4?
- Why is the data cleanup (removing data points) necessary?
- The accuracy comparison between the off-chip and on-chip baseline is useful for this proof-of-concept. But additionally, how fast is the on-chip learning? Given the desire for neuromorphic computing to be able to perform on-chip learning additional performance details will be necessary for publication.

---

### Decision · Program_Chairs · 2024-10-10

Accept (Poster)